# OpenReview forum: "Guaranteed Jailbreaking Defense via Disrupt-and-Rectify Smoothing"
_ICLR.cc/2026/Conference — Submitted to ICLR 2026_

### Official Review · Reviewer_nGLK · 2025-10-27

**Soundness:** 2
**Presentation:** 2
**Contribution:** 2
**Rating:** 4
**Confidence:** 3

**Summary:**

This paper proposes Disrupt-and-Rectify Smoothing (DR-Smoothing) defense against jailbreak attacks on large language models. DR-Smoothing introduce a two-stage smoothing scheme: first, disrupt the input prompt through randomized perturbations, then rectify it using spell-checking and paraphrasing. Aggregated responses are combined via majority voting to produce the final output. The method generalizes existing smoothing-based defenses by restoring out-of-distribution perturbed prompts to in-distribution form. Empirical evaluations across multiple LLMs show DR-Smoothing outperforms prior defenses.

**Strengths:**

1. The paper introduces the *Disrupt-and-Rectify* paradigm, extending randomized smoothing theory from adversarial robustness to the realm of jailbreaking defense.
2. The writing is clear and well-structured, with effective use of figures, pseudocode, and tables to illustrate the workflow.
3. The paper successfully bridges theoretical analysis and practical implementation, demonstrating how DR-Smoothing influences the Lipschitz behavior of perturbation responses and interprets empirical results through mathematical foundations.

**Weaknesses:**

1. Although the paper includes an efficiency analysis, DR-Smoothing inherits the common drawback of smoothing-based methods — it requires multiple LLM queries per input, which may incur substantial computational and latency overhead. This makes the approach less practical for real-time or API-limited deployment scenarios.
2. Limited Methodological Novelty.  The proposed two-stage design, while conceptually neat, integrates existing techniques (e.g., spell-checking and paraphrasing) within the disruption and rectification modules.
3. The experimental evaluation is limited to AdvBench and two jailbreak types (GCG and PAIR). The absence of broader testing on more diverse or state-of-the-art jailbreak scenarios weakens the empirical generality of the claims.

**Questions:**

1. How does DR-Smoothing perform under multi-turn or context-dependent jailbreaks, where harmful intent is distributed across several dialogue turns rather than a single prompt?
2. The two-stage Disrupt-and-Rectify process is central to the proposed method. Could the authors provide visual or quantitative analyses (e.g., embedding visualization) to illustrate how the rectification stage transforms the prompt distribution and contributes to defense success?

---

> ### Author Response · Authors · 2025-11-21
>
> Dear Reviewer nGLK,
>
> We thank the reviewer for their constructive comments and for the time spent reviewing our paper. We have carefully considered all points and address them below.
>
> > [Q1]How does DR-Smoothing perform under multi-turn or context-dependent jailbreaks, where harmful intent is distributed across several dialogue turns rather than a single prompt?
>
> Thanks for the comments. We conducted additional experiments to evaluate our method against the multi-turn jailbreak attack Crescendo. On both Vicuna and Llama models, our approach outperforms SmoothLLM, demonstrating that it retains strong defensive capability even under multi-turn jailbreak scenarios.
>
> | Defense Methods | Vicuna | Llama |
> | --------------- | ------ | ----- |
> | None            | 32     | 23    |
> | SmoothLLM       | 32     | 21    |
> | Ours            | 29     | 18    |
>
>
>
> > [Q2]The two-stage Disrupt-and-Rectify process is central to the proposed method. Could the authors provide visual or quantitative analyses (e.g., embedding visualization) to illustrate how the rectification stage transforms the prompt distribution and contributes to defense success?
>
> We randomly sample several **successful-defense** pairs (cases in which our method successfully defends against a jailbreak prompt: the blue dot denotes the original jailbreak prompt, while the blue cross represents the prompt after disrupt-and-rectify) and **failed-defense** pairs (cases in which our method fails to defend: the red dot denotes the original jailbreak prompt, and the red cross represents the prompt after disrupt-and-rectify), along with a set of **benign prompts**.
>
> The figure has been incorporated into the revised manuscript. From the figure, we observe the following:
> (1) Our defense consistently shifts prompt representations into a particular region of the embedding space, regardless of whether the defense ultimately succeeds or fails.
> (2) The movement distance for successful-defense cases is noticeably smaller than that for failed-defense cases.
>
> > [W3]The experimental evaluation is limited to AdvBench and two jailbreak types (GCG and PAIR). The absence of broader testing on more diverse or state-of-the-art jailbreak scenarios weakens the empirical generality of the claims.
>
> We thank the reviewer for this valuable comment. In response, we have expanded our evaluation to include two additional and more advanced attacks, TAP and AutoDAN-Turbo, both of which represent substantially stronger jailbreak strategies. The updated results are shown below:
>
> | Attack Methods | Defense Methods | Vicuna | Llama |
> | -------------- | --------------- | ------ | ----- |
> | TAP            | None            | 52     | 11    |
> |                | SmoothLLM       | 46     | 2     |
> |                | Ours            | 40     | 2     |
> | Autodan-Turbo  | None            | 75     | 31    |
> |                | SmoothLLM       | 52     | 24    |
> |                | Ours            | 46     | 21    |
>
> These results confirm that DR-Smoothing maintains strong performance even under more challenging and modern jailbreak scenarios, addressing the reviewer’s concern about empirical generality.

---

### Official Review · Reviewer_XAM2 · 2025-10-27

**Soundness:** 3
**Presentation:** 3
**Contribution:** 2
**Rating:** 2
**Confidence:** 4

**Summary:**

This paper proposes a jailbreak defence strategy, Disrupt and Rectify Smoothing. At a core level, in this defence, a input prompt is first perturbed (in a similar manner to SmoothLLM), and then undergoes a rectification step (spell check, paraphrasing, etc). This has two advantages in the paper. First, the prompt is cast into a form that for benign queries the LLM is better able to understand and thus suffers from a lower false positive rate. Second, the increased modifications from the paraphrasing stage can further degrade the jailbreak quality.

**Strengths:**

The paper tackles a important and timely problem, particularly with the growing use of LLMs not just as chatbots, but within more sensitive agentic workflows.

On the benchmarks and models tested, there is a improvement in performance compared to several compared to defences. In particular the authors use the IntructionFollowing metric to show the small impact on performance of the defence. It also seems from their results that the operations of rectification and smoothing do not cancel each other out (although in theory possible) when used - at least against non-defence tailored attacks.

**Weaknesses:**

The evaluation is relatively small - only GCG and PAIR attacks are considered. Many attacks have been developed since, for example TAP being a stronger iteration of PAIR, AutoDAN, Crescendo, MadMax, etc are all valid attacks.

The adaptive attack consideration is relativity lightweight - in fact the adaptive attack considered in Table 2 performs worse against DR-Smoothing than the results in Table 1. Likewise, there is no discussion of how an attack may try and exploit or bypass the rectification process directly seeing as it is paraphrased using an LLM.

The novelty is somewhat modest, as this defence builds directly on SmoothLLM/SemanticSmooth with a conceptually straightforward extension with known methods. In particular, SemanticSmooth already introduced the idea of Spellchecking and Paraphrasing in the defence pipleine. The paper further compares their metrics only against SemanticSmooth's uniform sampling technique, describing it as a fairer comparison in Line 376-377. This proves problematic as excluding the stronger SemanticSmooth-Policy in the analysis dilutes the contribution of the current approach.

Finally, the rectification step could use further discussion: there is little detail about the LLM used, prompt setup, or how rephrasing quality may impact the performance of the defence.

**Questions:**

Why was SemanticSmooth-policy excluded from the experimental comparison? Given the original paper reported that as the strongest variant of the defence.

---

> ### Author Response · Authors · 2025-11-21
> **1/2**
>
> Dear Reviewer XAM2,
>
> We thank the reviewer for their constructive comments and for the time spent reviewing our paper. We have carefully considered all points and address them below.
>
> > [W1]The evaluation is relatively small - only GCG and PAIR attacks are considered. Many attacks have been developed since, for example TAP being a stronger iteration of PAIR, AutoDAN, Crescendo, MadMax, etc are all valid attacks.
>
> Thanks for the comments. We conducted experiments to evaluate our method against recent jailbreak attacks, including TAP, AutoDAN-Turbo, and Crescendo. Note that AutoDAN-Turbo is a state-of-the-art extension of AutoDAN. On both Vicuna and Llama models, our approach consistently outperforms SmoothLLM, demonstrating the clear advantage of our disrupt-and-rectify scheme.
>
> | Attack Methods | Defense Methods | Vicuna | Llama |
> | -------------- | --------------- | ------ | ----- |
> | TAP            | None            | 52     | 11    |
> |                | SmoothLLM       | 46     | 2     |
> |                | Ours            | 40     | 2     |
> | Autodan-Turbo  | None            | 75     | 31    |
> |                | SmoothLLM       | 52     | 24    |
> |                | Ours            | 46     | 21    |
> | Crescendo      | None            | 32     | 23    |
> |                | SmoothLLM       | 32     | 21    |
> |                | Ours            | 29     | 18    |
>
>
>
> > [W2]The adaptive attack consideration is relativity lightweight - in fact the adaptive attack considered in Table 2 performs worse against DR-Smoothing than the results in Table 1. Likewise, there is no discussion of how an attack may try and exploit or bypass the rectification process directly seeing as it is paraphrased using an LLM.
>
> We thank the reviewer for the thoughtful observation.
>
> In Table 2, the evaluation metric is consistent with Table 1, as both employ ASR. A higher ASR indicates stronger defense performance. For example, using Vicuna as an example: under the established PAIR attack, our DR-Smoothing achieves an ASR of 36\%, whereas under the adaptive PAIR attack, the ASR slightly increases to 40\%, indicating that the adaptive PAIR attack is indeed stronger than the established version.
>
>
>
> > [W4]Finally, the rectification step could use further discussion: there is little detail about the LLM used, prompt setup, or how rephrasing quality may impact the performance of the defence.
>
> We thank the reviewer for highlighting this point. We use **gpt-3.5-turbo-1106** for paraphrasing. The exact prompt is:
>
> ```typescript
> In this task, you will receive an English instruction. Your goal is to paraphrase it.
>
> Follow the guidelines:
> 1. Paraphrase the instruction one sentence by one sentence.
> 2. Ensure the paraphrase retains the original intent of the instruction.
> 3. Do not omit any sentence from the original instruction.
>
> Input: {input_text}
>
> Now paraphrase the instruction in the input. Generate the paraphrase in a JSON object. The JSON object has only one key: 'paraphrase'.
> ```
>
> We will add additional discussion about paraphrasing quality and its potential impact on defense behavior in the final version.

---

> > ### Author Response · Authors · 2025-11-21
> > **2/2**
> >
> > > [Q1]Why was SemanticSmooth-policy excluded from the experimental comparison? Given the original paper reported that as the strongest variant of the defence.
> >
> > We sincerely thank the reviewer for this question.
> >
> > We compare our approach with **SemanticSmooth-uniform**, rather than **SemanticSmooth-policy** to ensure a fair evaluation. Both our method and SemanticSmooth-uniform employ a uniform random sampling strategy to select a single disruption operation, making the comparison directly aligned.
> >
> > It is worth noting that SemanticSmooth relies on SEVEN disruption operations (Spellcheck, Verbtense, Synonym, Paraphrase, Translate, Summarize, and Format). According to Table 2 of the SemanticSmooth paper, Summarize and Format are substantially more effective than the other five. By contrast, our approach employs only two simple operations—Spellcheck and Paraphrase—yet still outperforms SemanticSmooth even when it uses all seven operations. Moreover, when compared specifically against SemanticSmooth’s Spellcheck or Paraphrase variants—the fairer comparison—our method surpasses them by an even larger margin. All of these results clearly demonstrate the advantage of our disrupt-and-rectify scheme.
> >
> > We acknowledge that **SemanticSmooth-policy** is more effective than **SemanticSmooth-uniform**, as it leverages an adaptive mechanism to select the most appropriate disruption operation for each jailbreak method and input query. However, this adaptivity inevitably incurs additional computational overhead, such as training a policy network via reinforcement learning. We contacted the SemanticSmooth authors to request their code but received no response. In our own attempts, we found that training the policy network was highly unstable and inefficient, as collecting successful jailbreak samples for reinforcement learning is nontrivial. Nonetheless, we plan to explore a more efficient adaptive selection mechanism in future work, as noted in Line 377.

---

### Official Review · Reviewer_9pcy · 2025-10-30

**Soundness:** 3
**Presentation:** 3
**Contribution:** 3
**Rating:** 8
**Confidence:** 3

**Summary:**

This paper tackles the problem of out-of-distribution perturbed prompts causing unpredictable LLM behavior in smoothing defenses (e.g., SmoothLLM). It proposes DR-Smoothing, a two-stage (disrupt-rectify) smoothing method. Key results: It outperforms SOTA baselines, reducing Vicuna’s GCG ASR to 3.4% (vs SmoothLLM’s 8.6%) and maintaining higher InstructionFollow accuracy, with theoretical DSP bounds.

**Strengths:**

1. Rigorous theoretical foundation: Unlike empirical-only baselines, it derives tight bounds for Defense Success Probability (DSP) and disruption strength requirements (e.g., q ≥ 1/(2L)(1+√(2/N)log(1/ε))), providing mathematical guarantees for defense effectiveness.
2. Two-stage prompt processing: The rectification module (spell-check + paraphrasing) restores out-of-distribution disrupted prompts to in-distribution form, avoiding unpredictable LLM behavior seen in SmoothLLM (e.g., rectified prompts reduce gibberish-induced errors).
 3. Adaptive attack resilience: It defends against adaptive PAIR attacks (e.g., Vicuna’s adaptive ASR stays low vs Perplexity Filter’s 98%), showing robustness to adversary-aware attacks.

**Weaknesses:**

1. Random disruption operation selection: It randomly chooses character/word-level disruptions; adaptive selection (e.g., character-level for GPT, word-level for PAIR) could optimize efficiency—adding a dynamic selector based on attack type would improve performance.
2. Scalability issues: N=10 (standard setting) increases runtime to 7.7s (vs baseline 0.7s); optimizing N (e.g., N=3 for lightweight scenarios) without ASR loss is unaddressed, limiting deployment in low-latency systems.
3. Limited model scale testing: It only evaluates 7B models (Llama-2-7B, Vicuna-7B); larger models (e.g., 13B/70B) are untested—verifying on larger models could confirm scalability.

**Questions:**

Please refer to the weaknesses above.

---

> ### Author Response · Authors · 2025-11-21
>
> Dear Reviewer  9pcy,
>
> We thank the reviewer for their constructive comments and for the time spent reviewing our paper. We have carefully considered all points and address them below.
>
> > [W1]Random disruption operation selection: It randomly chooses character/word-level disruptions; adaptive selection (e.g., character-level for GPT, word-level for PAIR) could optimize efficiency—adding a dynamic selector based on attack type would improve performance.
>
> Thanks for the idea, we acknowledge that adaptive disruption selection can further enhance defense performance, and we plan to integrate such a scheme into our approach in future work (Line 377). Note that SemanticSmooth-policy employs a reinforcement-learning–based mechanism that requires high-quality reward modeling, whereas we are trying to develop a lightweight yet efficient solution.
>
>
>
> > [W3]Limited model scale testing: It only evaluates 7B models (Llama-2-7B, Vicuna-7B); larger models (e.g., 13B/70B) are untested—verifying on larger models could confirm scalability.
>
> Thanks for the comment. We will evaluate our approach on larger models.

---

### Official Review · Reviewer_m5dK · 2025-10-30

**Soundness:** 1
**Presentation:** 2
**Contribution:** 1
**Rating:** 0
**Confidence:** 4

**Summary:**

This paper proposes DR-Smoothing, a defense algorithm consisting of random disruption and recitifacation to achieve certified robustness.

**Strengths:**

1. The authors conducted experiments across several models and datasets.

**Weaknesses:**

1. The theoretical proof is meaningless and offers no new insight into the jailbreak problem. I cannot see why the proof supports the claim that the method is certified. Specifically, the theory states that \alpha(q) should be above certain threshold. However, the authors fail to provide practical guidelines on how to achieve the threshold and it is not known whether such assumption will hold in practise.

2. The experiments are all conducted on out-of-datad benchmarks with out-of-dated LLMs. DR-Smoothing does not offer superior balance between robustness and accuracy. For example, the method reduces accuracy for 7-8% compared to the baseline, which is significant.

3. What is the paraphrasing function? How is it implemented?

4. The additional inference cost of MV and paraphrasing is huge. This defense is neither theoretically sound nor practically useful.

To summarize my points, the proposed method fails to provide any advancement compared to existing defenses based on random smoothing both empically and theoretically. The technical contribution is limited to trivial re-writing and voting, and the theory is merely a direct application of Hoeffding’s inequality. The scope of the experiments is small and there are important details left unclear.

**Questions:**

What is the paraphrasing function? How is it implemented?

---

> ### Author Response · Authors · 2025-11-21
>
> Dear Reviewer m5dK,
>
> We thank the reviewer for their constructive comments and for the time spent reviewing our paper. We have carefully considered all points and address them below.
>
> > [W1]The theoretical proof is meaningless and offers no new insight into the jailbreak problem. I cannot see why the proof supports the claim that the method is certified. Specifically, the theory states that \alpha(q) should be above certain threshold. However, the authors fail to provide practical guidelines on how to achieve the threshold and it is not known whether such assumption will hold in practise.
>
> We thank the reviewer for raising this important theoretical concern.
>
> Our theory explains why smoothing strategies—not only our DR-Smoothing but all smoothing-based approaches including SmoothLLM—are inherently beneficial for jailbreak defense. The essence of these methods is to inject noise and perform inference N times. Compared with a single forward pass, conducting N inferences substantially increases the probability of successful defense. Our theoretical result shows that “increasing N can significantly reduce the threshold/requirement on alpha(q)”, meaning that the smoothing of performing N times inferences is critical.
>
>  It is important to note that alpha(q) depends on many factors, including the jailbreak method, the target LLM, and even the specific input prompt. Therefore, our approach does not attempt to directly reduce alpha. Instead, we focus on relaxing the threshold requirement on alpha by increasing N.
>
> Although the SmoothLLM paper offers some preliminary analysis, it only provides a way to compute the defense success probability (DSP), without deriving the threshold required to achieve any DSP > (1- epsilon), nor offering an explicit characterization of how this threshold scales with the number of inference rounds N. In contrast, our theory is considerably more clear and elegant. It explicitly characterizes the mechanism through which conducting N time inferences reduce the requirement on alpha, thereby revealing the fundamental rationale behind the robustness gains of smoothing-based methods.
>
>
>
> > [W2]The experiments are all conducted on out-of-datad benchmarks with out-of-dated LLMs. DR-Smoothing does not offer superior balance between robustness and accuracy. For example, the method reduces accuracy for 7-8% compared to the baseline, which is significant.
>
> We thank the reviewer for pointing this out.
>
> Achieving a balance between defense effectiveness and model utility is a long-standing challenge in jailbreak defense. Although our method inevitably introduces some utility degradation, it still surpasses prior approaches by a substantial margin. For instance, on Llama-2, our approach attains a utility score of 28.7%, compared with 17.0% for SmoothLLM and 21.3% for SemanticSmooth.
>
>
>
> > [W3]\[Q1]What is the paraphrasing function? How is it implemented?
>
> We thank the reviewer for the question. We use **gpt-3.5-turbo-1106** with the following prompt:
>
> ```typescript
> In this task, you will receive an English instruction. Your goal is to paraphrase it.
>
> Follow the guidelines:
> 1. Paraphrase the instruction one sentence by one sentence.
> 2. Ensure the paraphrase retains the original intent of the instruction.
> 3. Do not omit any sentence from the original instruction.
>
> Input: {input_text}
>
> Now paraphrase the instruction in the input. Generate the paraphrase in a JSON object. The JSON object has only one key: 'paraphrase'.
> ```

---

> ### Comment · Reviewer_m5dK · 2025-11-26
>
> I appreciate the authors' for their effort in preparing the response, whichi is helpful. However, my concerns remain and I will keep my initial rating of 0.
>
> 1. The theory is a naive application of the concentration inequality, which is trivial. It is straightfoward that you perturb the input for enough times and you will have certain probability to pass. But what about the cost? And how is your theory different from so many existing works in random smoothing.
>
> 2. The experimental results are weak. The natural accuracy is significantly hurt.

---

> > ### Author Response · Authors · 2025-11-27
> > **LLM jailbreaking defense differs fundamentally from adversarial robustness**
> >
> > Thanks for your feedback,
> >
> > To the best of my knowledge, only two prior works attempt to introduce random-smoothing techniques into the LLM domain—SmoothLLM and SemanticSmooth. SemanticSmooth provides no theoretical analysis, while SmoothLLM offers only preliminary analysis. In particular, its Proposition 3.3 does not establish an **explicit relationship** between the defense success probability (DSP) and the number of smoothing samples N, and its notion of k-unstable in Eq. (3.4) is difficult to compute in practice. In contrast, our work derives an **explicit bound** for the DSP and its **dependence** on N.
> >
> > Furthermore, prior random-smoothing research primarily targets **small models**, particularly in **image-classification** settings, where smoothing is applied to defend against **pixel-space adversarial perturbations**. However, unlike small vision classifiers, LLMs are **generative models**, and **LLM jailbreaking defense differs fundamentally from adversarial robustness** in its objectives, methodologies, and theoretical underpinnings. Notably, smoothing-based defenses remain largely unexplored in the context of LLM jailbreak attacks, and existing results developed for adversarial robustness cannot be directly applied to jailbreak defense. Moreover, jailbreak attacks are substantially more challenging to defend than pixel-level adversarial attacks, making the extension of smoothing techniques to this domain far from straightforward.
> >
> > In the LLM safety domain, the development of jailbreak defenses has lagged behind the rapid progress of jailbreak attack techniques, and we hope that our work contributes meaningfully to strengthening LLM jailbreak defense.

---

### Author Response · Authors · 2025-12-02
**Summary of Reviewer Suggestion and Rebuttal Statu**

Dear Area Chair,

First, we would like to express our sincere gratitude for your time and effort in handling our submission. To facilitate your efficient evaluation of our work, we provide a summary of the current reviewer status and the major improvements made during the rebuttal.

**1. Reviewer XAM2 (Initial Score: 2) raises the concern: “Why was SemanticSmooth-policy excluded from the experimental comparison?”**

Our Response:
We compare our method against SemanticSmooth-uniform rather than SemanticSmooth-policy to ensure a **fair and directly aligned evaluation**. Both our approach and SemanticSmooth-uniform rely on uniform random sampling to choose a single disruption operation, making their mechanisms comparable on equal footing.

In contrast, SemanticSmooth-policy introduces substantial additional computational overhead, including training a policy network via reinforcement learning and sampling successful jailbreak prompts—an inherently unstable and resource-intensive process. Since our focus is on evaluating disruption-based defenses under comparable computational budgets, SemanticSmooth-uniform provides the most appropriate baseline. Nevertheless, as noted in Line 377 of our manuscript, we plan to investigate more efficient adaptive selection strategies in future work.

**2. Reviewer m5dK (Initial Score: 0) raises the concern that “The theoretical proof is meaningless and offers no new insight into the jailbreak problem.”**

Our Response:
To the best of our knowledge, our work presents the first comprehensive theoretical analysis of applying smoothing techniques to the LLM jailbreak defense setting. While numerous studies investigate smoothing-based certification for **adversarial attacks**, LLM jailbreak defense differs fundamentally from adversarial robustness in its objectives, methodologies, and theoretical underpinnings. Existing smoothing theories developed for image classifiers or small models cannot be directly applied to the generative and discrete nature of LLM jailbreak scenarios.

Our analysis is tailored to this new problem setting and **establishes explicit bounds and guarantees** that, to our knowledge, have not been explored in prior jailbreak-defense literature. We believe this provides meaningful theoretical insight into strengthening LLM jailbreak defense and opens a principled direction for future research in this underexplored but increasingly important area.

**3. Reviewer 9pcy (Initial Score: 8) and Reviewer nGLK (Initial Score: 4) requested additional details about our approach as well as more comprehensive comparisons with related methods.**

Our Response:
We have conducted further experiments and incorporated additional comparative analyses that more clearly demonstrate the advantages of our method. We have also expanded the methodological description to provide greater clarity and completeness.

Dear Area Chair, we understand that taking over the assignment at this stage requires significant energy, and we deeply appreciate your dedication to the review process.

---

### Meta-Review · Area_Chair_mZM2 · 2026-01-03

**Summary:**

- DR-Smoothing has high time overhead: Since DR-Smoothing includes both a perturbation stage and a response stage, its time overhead is relatively high.
 - DR-Smoothing significantly affects model general utility: The authors evaluate utility on only one benchmark, InstructionFollow, and the performance drop compared to the original model is relatively large.
 - Lack of methodological novelty: DR-Smoothing is not substantially different from SmoothLLM.

**Reviewer Concerns:**

## Reviewer m5dK

 - Theoretical proof is meaningless: Although the reviewer still believes the theoretical proof is meaningless, I think it nevertheless provides some benefit.
 - DR-Smoothing does not offer a superior balance between robustness and accuracy: Not resolved. The authors evaluate utility on only one benchmark, InstructionFollow, and the performance drops significantly compared to the original model.
 - Paraphrasing function: Resolved. The authors provide this function.
 - The additional inference cost of MV and paraphrasing is huge: Not resolved. The authors did not respond.

## Reviewer 9pcy

 - Random disruption operation selection: Not resolved. The authors did not add relevant experiments.
 - Scalability issues: Not resolved. The authors did not respond.
 - Limited model scale testing: Not resolved. The authors did not add relevant experiments.

## Reviewer XAM2

 - The evaluation is relatively small: Resolved. The authors added relevant experiments.
 - The adaptive attack consideration is relatively lightweight: Resolved. The authors explained the results.
 - The novelty is somewhat modest: Not resolved. The authors did not respond.
 - The rectification step could use further discussion: Partially resolved. The authors provided the model type used for rewriting.

## Reviewer nGLK

 - DR-Smoothing may incur substantial computational and latency overhead: Not resolved. The authors did not respond.
 - Limited methodological novelty: Not resolved. The authors did not respond.
 - The experimental evaluation is limited to AdvBench and two jailbreak types: Resolved. The authors added two jailbreak methods.
 - How does DR-Smoothing perform under multi-turn or context-dependent jailbreaks: Partially resolved. The authors added one jailbreak method.
 - Could the authors provide visual or quantitative analyses: The authors provided them.

**Reviewer Scores:**

## reviewer GUAA

 - Score remains 0

## reviewer RBU3

 - Score remains 8

## reviewer b1Lx

 - Score remains 2

Scores cannot be changed due to system reasons.
## reviewer FHc7

 - Score remains 2

---

### Decision · Program_Chairs · 2026-01-26

Reject